# Gene Editing: The Regulatory Perspective

Sarfaraz K. Niazi

College of Pharmacy, University of Illinois, Chicago, IL 60612, USA; sniazi3@uic.edu or niazi@niazi.com

**Definition:** Gene or genome editing, often known as GE, is a technique utilized to modify, eliminate, or substitute a mutated gene at the DNA level. It serves as a valuable tool in the field of genetic manipulation. Gene therapy (GT) is a therapeutic approach that aims to correct mutations by delivering a functional gene copy into the body. In contrast, the mutated gene remains in the genome. It is considered a form of medical intervention. No approval has been granted for any product manufactured by GE, in contrast to the approval of 22 medications produced by GT. These GT products are priced at millions of US dollars each dose. The Food and Drug Administration (FDA) has recently implemented a guideline about gene editing, which aims to facilitate the expedited creation of genetically engineered (GE) goods. However, the FDA must provide further elucidation and necessary revisions to enhance the rationality of this guideline.

**Keywords:** gene editing; gene therapy; FDA; regulatory approval

## 1. Introduction

Mutations cause human evolution and a myriad of diseases. Treatment for roughly a billion people with faulty, disease-causing genes was impossible before gene modification technologies. While the FDA has approved 22 gene therapy (GT) products [1], and the EMA [2] has approved 13 products (labeled as Advanced Therapy Medicinal Products) (ATMPs), neither agency has approved any GE product. However, many are under development [3]. Thus, the developers should improve their understanding of the scientific, technical, and regulatory perspectives based on the lessons learned from the GT products to reduce the cost and time of bringing them to patients. This is the primary focus of this paper.

The cost of GT and GE products to patients is exorbitant [4]. However, there are many creative possibilities to reduce this high cost and the long time it takes to secure regulatory approval. These include outsourcing the work, partnering with academic institutions, using special IND approvals, working closely with regulatory agencies, and questioning the listed testing requirements. In the future, this category will likely include GE products as well. In addition, a better understanding of individual variability and next-generation sequencing (NGS) technologies to discover new uncommon genetic illnesses have made individualized therapies more practical.

Regulators can take additional steps to support the uniformity of off-target (and on-target) effect measurement, such as putting the best practices for sample handling and analysis into place, as well as quality control checks. Because they are still being developed, methods for identifying on- and off-target effects are not specified in the EU rules for the quality, non-clinical requirements, and clinical requirements of genetically modified cells, for instance [5].

The impact of the in vivo cellular environment on gene editing efficiency is very important. This poses challenges in accurately forecasting the clinical outcomes of gene editing treatments in people, particularly when relying on non-clinical efficacy models. Given the potential scenario where gene editing treatments may not necessitate or lack pertinent non-clinical efficacy models, engaging in comprehensive inquiry and endeavors

to establish such relevant ones is imperative. These efforts should be thoroughly deliberated and requested individually during regulatory interactions.

Japan's PMDA produced a Guideline on Assuring the Quality and Safety of Gene Therapy Products (not gene editing-specific). This causes scientific and political concerns. To keep potentially curative ATMPs orphan, they must have a significant advantage over authorized ATMPs. Most ex vivo GE products are GTMPs or cell treatments, meeting ATMP criteria. The FDA classifies all gene editing products (in vivo and ex vivo) as gene therapy in the US. For scientific reasons, clinical efficacy data may not be available to verify superiority, leaving clinical safety or non-clinical data to support a significant benefit claim.

## 2. CMC Considerations

Based on pipeline evaluations and clinical success rates, the FDA plans to approve dozens of cell and gene therapy items yearly by 2025 [6]. In anticipation, the FDA has been highly proactive in bringing regulatory guidance for Tissue and Advanced Therapies products by announcing an ambitious plan in 2022 that included guidance documents [7] that included neurodegenerative diseases, gene editing, chimeric antigen receptor (car) t cells, human cells, tissues, and cellular and tissue-based products, regenerative medicine, manufacturing changes and comparability for human cellular and gene therapy products, early phase clinical trials, and several other topics.

Of significance is the guideline issued in March 2022, providing guidance and recommendations on developing GE of human somatic cells, more particularly to evaluate the GE products' quality and safety as per Title 21 of the Code of Federal Regulations 312.23, information that must be included in an IND application (21 CFR 312.23). This includes evaluating preclinical safety, designing clinical trials, manufacturing, testing, and designing products.

The FDA guidance has repeatedly emphasized safety, specificity, and function analysis and recognized the difficulties in developing these products; while most of these descriptions are related to gene therapy (G)T products, these also apply to GE products.

Despite the difference in the GE and GTs, the FDA applies the identical Chemistry, Manufacturing, and Control (CMC) requirements to other GE products, both being treated as new biological drugs. Thus, all requirements applied to new BLA approvals apply to GE, regardless of whether it is personalized medicine or a commercial product for distribution [8].

However, the FDA promotes the need for increased mechanistic understanding, enhanced manufacturing capabilities, and new tools to achieve the potential of precision medicine and tailored therapies fully. The FDA is investigating new technologies (such as omics) to improve disease diagnosis, prognosis, and treatment advancements. The FDA has also created precisionFDA [8], a cloud-based community research and development platform that enables users worldwide to exchange GE data and tools for testing, piloting, and validating old and new bioinformatics methods for next-generation sequencing (NGS) processing.

## 3. Drug Substance

The GE components should be improved to minimize the possibility of off-target gene modification. Depending on the specific GE technology, the optimization process can be performed on either the editor or the specific components. The guide RNA is a genetic engineering component that can be modified to inhibit degradation. The IND needs to include a thorough description of the optimization technique.

This stage of development is responsible for most of the cost and time to qualify a product for testing. Off-target events are either related to the variability of the active components or innate responses. While several AI methodologies can ascertain the structural variability, the innate response will become more understandable once we have sufficient data. Of course, this would increase the risks and side effects of gene editing, but new technologies and further GE advancements might eventually get around these obstacles.

GE components can either affect cells ex vivo or be delivered in vivo utilizing nanoparticles, plasmids, or viral vectors. The GE components are considered active medicinal ingredients or therapeutic compounds when given as nanoparticles in vivo and incorporated into DNA, RNA, or protein.

GE components are often regarded as a drug product (DP) in its final formulation for in vivo injection. The designated term for the plasmid or vector encoding the genetic engineering (GE) component is the Delivery Platform (DP), when the GE components are expressed in vivo through the direct administration of plasmids or vectors. Furthermore, the quality of GE components plays a vital role in producing the final product when used for ex vivo cell modification.

The Investigational New Drug (IND) application should describe how the GE tool is manufactured in a GMP environment and tested [9,10]. All conditions of pharmaceutical products apply to GE tools as well.

The ethics of testing GE tools is also complicated since this designation and its requirements create significant ethical issues; administering a gene-altering therapy in healthy subjects should not be allowed. The same applies to Phase 2 studies. In addition, there is no guidance about personalized medicines that could not be tested in any other population except the patient for which it is created. Developers should prepare a robust safety profile from animal testing to convince the FDA that the testing should be conducted in patients, even at a small scale; studies with one or two subjects are familiar.

Stability testing is required when a product is commercially distributed with a listed shelf-life; only stability data during the testing period are required for IND purposes. Products manufactured for immediate administration should only be required to demonstrate meeting the release specification during administration. Developers can also choose the option of storage requirements known to provide a stable environment, such as $-70°C$; when Pfizer's COVID-19 vaccine was approved, it was supposed to be stored at a very low temperature to avoid conducting complete stability testing; it was later changed. Some gene and cell therapy products must be held at $-70$ to $-190\ °C$, at which temperature and stability testing are optional for IND purposes.

## 4. Drug Product

Clinical INDs are widely used to customize gene therapy products in hospitals and academic units; INDs aim to ensure that the product has reasonable promise to demonstrate the expected outcome, is safe, and that human testing is not abused. While commercial INDs include all the information listed above, given that these studies are conducted in a few qualified patients, basic safety should suffice to secure IND approvals. In addition, developers may establish academic institutes [11] and hospital relationships to take a faster route to IND approvals.

Many of the requirements to address safety issues are not likely to be available when filing the IND; safety concerns due to the manufacturing process will still be unknown, and the same holds for the requirement of the efficiency of GE products. Sterility is, of course, required.

Test methods required to release a product must be validated. Still, for some tests specific to GE, only suitability can be demonstrated, mainly if it is used with a reference material or the reference product if it is a copy of an approved product.

The DP specifications are established for biological products based on the product character; there are no targeted final product attributes to be assessed; starting material specification will come from the supplier, and it should be acceptable to the FDA, provided the material claims cGMP grade.

With these technologies' ongoing improvements, Zinc-finger nucleases (ZFNs) and transcription activator-like effector nucleases (TALENs) and clustered regularly interspaced short palindromic repeats–bacterial RNA-guided endonuclease (CRISPR-Cas9), we have already begun using them in human clinical studies. However, while ex vivo GE results in highly effective cell therapies, correcting genetic diseases with ex vivo GE could be

more practical. Higher side effects like off-target editing, inefficiency, and the induction of negative immune reactions come at a cost.

The DP is the completed version of the plasmid/vector when used to express GE components in patients in vivo via a viral or plasmid vector. As a result, the IND needs to include a thorough explanation of plasmid/vector synthesis and testing.

Incorporating a comprehensive overview of the tests performed on the drug product (DP) and each nanoparticle component is imperative. It is essential to acknowledge that the testing process should encompass examinations to assess the extent to which each genetic engineering (GE) component has been integrated into the nanoparticles. Certain nanoparticles used to transport genetically modified components in vivo may be used for delivery.

The developers should seek clarification of what is considered a delivery device since the currently approved nanoparticles for COVID-19 are not treated as a device.

Developing potency assays for in vivo human GE DPs should assess the GE components' ability to perform the targeted molecular genetics and downstream biological modifications in target cells or tissues.

When discussing ex vivo-modified human gene-edited medicinal product manufacturing, process controls, and in-process testing for vital phases that may affect editing efficacy or specificity, are crucial. Ribonucleoprotein (RNP) production during CRISPR-mediated editing is vital and should be discussed in detail.

Acceptance standards or boundaries should also be offered and supported. The critical quality attributes (CQAs) are not known for new biological products as they are tested in the form presented; the quality attributes are then classified as critical once efficacy and safety are established. Therefore, there will be other options than defining these attributes at the pre-IND stage.

One notable distinction of ex vivo gene editing (GE) is the ability to conduct a broader range of tests. These tests include evaluating the efficiency of on-target editing, which involves characterizing the editing events at the intended site. Ex vivo GE may also measure off-target editing frequency, chromosomal rearrangements, residual GE components, and genome-edited cell counts. Stability testing of ex vivo-modified human GE donor cells can also track the number of edited cells or gene editing frequency. Many of these testing protocols are available from qualified CDMOs that can provide the service efficiently.

When creating potency testing for ex vivo-modified human GE DP, the assays will identify cell characteristics and the expected functional effects of genomic modifications from GE. Potency assays for a genome-edited CD34+ hematopoietic stem/progenitor cell product can quantify GE activity and function. Surrogate potency tests might be acceptable in some circumstances, but it is crucial that the data demonstrate a correlation between their results and the functional results of the GE. The relationship between surrogate potency and functional outcome is established once the clinical testing is completed. However, even then, the potency outcome rather than the relationship will require unnecessary extensive testing and exposure to patients.

Suppose the ex vivo-modified human gene-edited donor product (GE DP) is an allogeneic human cell product intended for several patients. In that case, further testing and establishing acceptability criteria may be required.

For instance, extra donor screening and testing may be necessary to meet the criteria for donor eligibility screening and testing. Additional requirements are not necessary, as already described for GT products. Also, each lot of the product is derived from different sources, making it impossible to establish acceptance criteria for the allogenic supply.

If the ex vivo-modified human GE DP is an allogeneic human cell product intended to treat several patients, additional testing and acceptance criteria may be needed. For instance, extra donor screening and testing may be necessary to meet the criteria for donor eligibility screening and testing. Listing these requirements as "may be warranted" leaves much uncertain; additional testing is also not defined. Is it over and above what is required for GT or other biological products? The testing should be relevant and only needed since,

at this stage, acceptance criteria are not established, and it is unclear what "stringent" means; these should be suitable and relevant.

Complex products that include many rounds of gene editing or the creation of different cell banks may necessitate additional testing during the manufacturing process and testing to ensure each batch's quality before release. Further explanation is needed regarding the phrases "additional" and "may be required," as it is essential to ensure that the appropriate testing is mandated.

## 5. Safety and Toxicity

Off-target GE refers to nonspecific and unintended genetic modifications that can arise based on whether the repair of the DNA takes place (nonhomologous end joining (NHEJ) or homologous recombination (HDR) responsible for site-specific modifications [12]; the latter is only active in dividing cells; thus, it is not applicable to the liver, neuron, muscle, eye, and blood stem cells. However, because HDR genes are found in all cells, they can be activated by giving the cells certain medications. The low rate of HDR in most cells is one factor that raises the possibility that genes will be disrupted rather than fixed in the first clinical application of CRISPR. Assume these complexes fail to bind at the target, frequently resulting from homologous sequences or mismatch tolerance. Then, they will cleave off-target DSB and result in non-specific genetic alterations [13], resulting in off-target effects like unintended point mutation [14] deletions [15], insertions, inversions, and translocations.

Although viral vectors, unlike their non-viral counterparts, are significantly more immunogenic and hazardous than their non-viral counterparts, this limits their therapeutic value. Adeno-associated virus (AAV) [16], pseudo-typed lentivirus, retrovirus [17,18] and baculovirus [19] are only a few examples of recombinant viruses used for gene transfer that have exhibited improved physicochemical and biological features after being covalently modified with PEG [20].

The Cas9 enzyme, intended to cut a specific DNA sequence, may also cause cuts in other genome regions, increasing the danger of mutations that increase cancer risk. The safety risk associated with CRISPR has received the most attention. However, CRISPR can be made more specific or tweaked to reduce off-target effects or increase the enzyme's capacity to exchange single DNA bases.

Another issue arises with the current use of CRISPR as an in vivo tool by introducing the Cas9 DNA into cells through a viral vector; even after Cas9 has made the desired cuts, cells will keep bringing it out for years, raising immune response to the enzyme, even if it is made highly specific. However, the risk is reduced if nonviral methods, such as lipid nanoparticles, are used, a rising technology.

Other difficulties include patients needing repeated treatments and the possibility that any gene-edited cells may eventually perish, depending on the condition. Less effectively than a single vector, two distinct viruses are frequently used to boost the maximum amount of DNA a viral vector can carry.

The safety and bioethical issues raised by somatic gene therapy using CRISPR/Cas9 are frequently like those raised when recombinant DNA technology and human gene therapy first became available [21]. The non-clinical challenge in safety and toxicity is thoroughly identifying off-target toxicities after gene editing. Off-target effects depend on the gene-editing tool's efficiency, delivery route, DNA target, cell type and stage of differentiation, chromatin structure, nuclease exposure length, and administration strategy (in vivo vs. ex vivo). In addition, editing errors can result in chromosomal translocations, insertions, deletions, and single nucleotide point mutations, all of which can be pathogenic to varying degrees.

Even when gene editing is successful, it can result in single nucleotide mistakes, added extra DNA, or "scarring" of the genome [22].

CRISPR-Cas-9 deletes or alters genes closer to the cut site in addition to DNA repair, primarily through error-prone nonhomologous end joining (NHEJ), increasing the risk of

pathogenic consequences that are close to the target. These can involve significant deletions or rearrangements that span thousands of base pairs.

In vivo, in vitro, and silico studies are sensitive and objective techniques to comprehend on and off-target. Biased approaches assess on/off-target effects utilizing knowledge of the gene editing product, while independent or "unbiased" approaches focus on the DNA (or other molecular targets). Among these unbiased methodologies, GUIDE-seq and CIRCLE-seq have complementary and sensitive nuclease activity definition methods [23,24].

## 6. Preclinical Studies

In non-clinical strategies for reducing patient risks, in simulating the best gene editor and distribution mechanism and increasing a gene editor's active duration there is still room for improvement.

Animal testing of gene editing products is a hotly contested issue. Nevertheless, animal modeling has been helpful in several situations. With the aid of CRISPR, numerous animal models with mutations that closely resemble the range of mutations found in people with Duchenne muscular dystrophy (DMD) can be created quickly. These models evaluate sequence-specific therapies like CRISPR, which reframe or skip DMD mutations to restore dystrophin expression [25].

Animal models are crucial for validating delivery systems inside living things. These animals also test potential medicines and detect side outcomes, including toxicity and immunogenicity. Regulatory authorities treat nearly all genome-editing therapeutics being advanced to the United States and the European Union clinic as needing target-indication-specific in-animal efficacy and safety studies. No matter the target cell, tissue type, or condition that needs to be treated, the SCGE initiative seeks to produce in vivo reporter systems that generally apply to diverse delivery and editing strategies. These reporters should be able to detect and quantify GE in the targeted tissue and edit events caused by non-specific dissemination to other tissues.

Preclinical examination of safety, efficacy, dose, and reagent distribution requires large animals while mice are appropriate for testing new delivery formulations due to their small size, low cost, and well-established utility. As an alternative, models are being developed to assess new delivery formulations' efficiency, specificity, and safety in wild-type and reporter-animal models, such as mouse reporter systems. Additionally, large animals like pigs and non-human primates can now be genetically modified accurately and efficiently thanks to engineered nucleases.

New non-invasive methods like total-body positron emission tomography (PET), magnetic particle imaging (MPI), and chemical exchange saturation transfer magnetic resonance aging (CEST MRI) are needed to track cells in vivo and develop new model organisms to measure editing effects [26].

Preclinical programs for human GE goods typically have long-term objectives similar to gene therapy products [27]. Dose ranges are established in typical phase 2 studies; given that GE products can only be tested in rare patients, conducting a complete dose-response analysis is impossible. However, developers should be able to determine a safe range of doses to start the trials with gradual increases if necessary.

These recommendations for the initial clinical dose selected, its escalation, schedule, and dosing frequently can only be estimates and need not be based on experimental evidence in humans; appropriate animal studies should allow sufficient data to make these projections.

There are limited routes of administration, and it is not feasible to try out several routes; based on the types of delivery systems, such as viral or LNP, the routes of administration and formulations are well-defined.

Toxicities of GE products cannot be identified based on the product design; the side effects of off-limit editing are the primary risks that can only be assessed in initial small-scale studies. Recent data on the possible immunogenicity and other side effects of formulation components such as polyethylene glycol should be addressed based on the frequency of

observed effects; this is particularly important since the tested population will be much smaller than the patient populations that demonstrate these side effects.

Preclinical in vitro and in vivo proof-of-concept (POC) investigations are recommended to assess the viability and scientific basis for deploying the investigational human GE product in a clinical study.

Animal testing models where genetic modifications are made offer a better solution. At the same time, these provide a scientific rationale; the lack of correlation between animal data and anticipated human response remains a barrier. An excellent scientific presentation to support the rationale should be provided.

When evaluating the efficacy of a genetically engineered (GE) product in terms of genomic change, it is essential to consider in vitro models that accurately represent the target cell type(s).

In several instances, the availability of such models may be limited. The recommendation effectively conveys this by emphasizing that their utilization "should be considered" rather than mandated. The experimental GE or species-specific surrogate product must elicit a physiological reaction in the selected animal species and models used for in vivo studies. Biological activity can be investigated in a species-specific environment, considering the differences in genome sequences between humans and animals. This approach allows for applying relevant findings to clinical products as needed.

The FDA is admitting that animal models may not be relevant, and these studies will be redundant. The FDA should provide more details about evaluating surrogate products.

The primary objective of preclinical safety research is to identify and assess the potential hazards of using the genetically engineered (GE) product. Possible toxicities could be associated with the delivery route, expression, genomic structural modification, and/or expression of genetically engineered (GE) components.

All these concerns were addressed earlier; the conclusion is that the developer should provide supportive data to ensure safety; newer technology, AI-based modeling, and NGS engagement can be helpful.

To the degree possible, safety evaluations should encompass off-target activities, chromosomal rearrangements, and their biological effects. Off-target projections are difficult to make and cannot be extrapolated from animal data; the developer should request IND approvals to determine these in a smaller population.

It is advisable to include elements of the planned clinical trial, such as the range of doses, route of administration, delivery method, frequency of dosing, and assessment criteria, in the preclinical safety studies conducted on an experimental genetically engineered product, to the degree that it is feasible. Furthermore, it is imperative that study designs possess adequate comprehensiveness to facilitate the detection, description, and measurement of potential local and systemic toxicities. This includes assessing the timing of their occurrence (immediate or delayed) and resolution and evaluating the impact of dosage levels on these outcomes.

These requirements are based on "to the extent possible", "sufficiently comprehensive", and "to the extent feasible". The developers should note this and submit to the FDA reasons for not submitting all the data requested above.

Biodistribution studies are carried out to describe the GE product's distribution, persistence, clearance, and any expressed GE components in vivo. Examining the biodistribution profile of the genetically altered sequence and the gene product's lifetime can reveal editing activity in target and non-target organs.

More clarification should be needed on how these tests should be conducted and whether animal data would suffice. This information will become available once the limited clinical testing is completed.

One specific recommendation suggests that it is important to assess the activity and safety of a genetically engineered (GE) product through definitive proof-of-concept (POC) studies, wherever possible. It may not be feasible for most GE products; this should be pointed out.

Due to variations in genomic sequences between humans and animals, employing a surrogate genetically engineered (GE) product in preclinical and safety studies may be necessary. This involves substituting the species-specific elements of the GE product, such as GE components, promoter(s), and transgene(s), with their human counterparts. This approach is adopted when administering the investigational human GE product, which would not provide informative results. Hence, it is recommended that developers establish the biological significance of the surrogate regarding the investigational human GE product and provide a scientific rationale for administering a surrogate GE product.

The FDA has yet to clarify these approaches that may not be practical, such as testing a surrogate product. Therefore, developers should seek clarification in the earliest meetings with the FDA.

The clinical cell source should be employed for the conclusive preclinical research for ex vivo-modified GE products. The scientific explanation for the cell source choice should be given in any study that uses an alternate cell source.

To prevent further justifications, it is advised to use clinical cell sources.

Based on the product development stage, each GE preclinical test lot should be described according to appropriate criteria. If applicable, this information will be essential for determining whether the product used in preclinical research can be compared to the clinical product.

Lots produced during preclinical stages should be at scale to avoid later justifications and validation; these are small-scale productions, so they should not bring a more significant cost burden.

The preclinical in vitro and in vivo POC studies are conducted for in-target and non-target cells, if applicable, to establish the specificity and efficacy of editing the functioning of the produced or repaired gene product (such as a protein or RNA). The durability of genomic change and its biological response, the editing efficiency needed to achieve the desired biological activity or therapeutic effect, and the effect of genetic diversity on target-population editing activity are examined in this study.

Preclinical studies should assess GE risk at on- and off-target loci and include the following:

- Identify off-target editing activity, including type, frequency, and location.
- It is advisable to use orthogonal techniques (such as in silico, biochemical, and cellular-based assays) that incorporate an objective genome-wide analysis to find potential off-target sites. Additionally, the study should utilize the target human cell type(s) from numerous donors whenever feasible.
- Appropriately sensitive procedures should be used to verify genuine off-target sites to detect low-frequency events. The study should also target human cells from many donors.
- To ensure the assay's quality, the results' interpretability, and their suitability for the intended use, appropriate controls should be included.
- An evaluation of the genomic integrity, which considers chromosomal rearrangements, significant insertions or deletions, the integration of exogenous DNA, and any potential for insertional mutagenesis or oncogenicity. This may involve checking for clonal growth and/or uncontrolled proliferation in ex vivo transformed cells.
- If possible, evaluate the biological effects of on- and off-target editing.
- Immunogenicity of the expressed gene product and GE components.
- Characterizing the expression and editing activity of GE components' kinetic profiles.
- Evaluation of the altered cells' viability and any selective survival advantage.
- Evaluation of the potential for unintentional germline modification.
- Preserving cell functionality after GE (such as progenitor cell differentiation capacity).

The extensive testing details provided by the FDA should be discussed before developing and testing a new product; many of these requirements will likely be waived in a comprehensive evaluation plan. Most of this work should be outsourced to a qualified contract development and manufacturing organization (CDMO).

## 7. Clinical Studies

Safety concerns for clinical testing include the effects of gene editing and the method used to deliver the gene-editing tool [28]. Human GE product clinical development programs address gene therapy product risks and GE risks like unanticipated effects of on- and off-target editing that may not be known during product administration.

The clinical trial design encompasses various essential components, such as the careful selection of patients, a reliable and efficient method of product administration based on data-driven dosage, a well-defined dosing schedule, and a comprehensive treatment plan. Additionally, the design contains thorough safety monitoring and a reasonable selection of endpoints. It is imperative to continuously monitor clinical trial participants who have been administered human gene editing (GE) products to evaluate the clinical safety of such interventions. Consequently, it is imperative for the Investigational New Drug (IND) application to provide a comprehensive description of the research design, the evaluation of adverse events (AEs), and the plans for subject follow-up. The overarching considerations for designing clinical trials with genetically engineered (GE) products align with those applicable to other cellular and gene therapy interventions [29].

Selecting an appropriate study population is crucial to optimize the advantages gained from the study while concurrently minimizing any potential hazards that may be posed to the participants. The study population is well selected based on the product mode of action (MOA), the study purpose, and product risks. Genetically modified (GE) human products may pose risks and offer doubtful benefits. As a result, only patients who have exhausted all other treatment options should be included in first-in-human trials for such products [29].

The risk of adverse events (AEs) associated with product delivery to target tissues must be reduced using established, secure, and efficient product delivery methods. When possible, prior clinical experience from analogous products, particularly genome-edited cellular or gene therapy products, should dictate delivery and dosing schedules. GE may have occurred on these goods [30].

Staggered subject enrolment, with a predetermined interval between product administration and subsequent subjects within and between cohorts, reduces the possible risk(s) associated with the GE product. The staggered period should be long enough to evaluate acute and subacute adverse effects (AEs) before giving more individuals the same dose or increasing it. The staggering interval should also consider how long a human GE product is anticipated to be active.

The proposed patient population size and its acceptable risk level for the GE product are considered when choosing the study cohort size. Other factors, including assessments of pharmacologic activity, tolerability, and feasibility, may also influence the choice of cohort size. Clinical trials of human GE goods must have a rigorous safety monitoring strategy, including a toxicity grading system and management plan. Monitoring off-target editing appropriately and evaluating the effects of undesired off-target and on-target editing repercussions are crucial.

In addition, the AEs connected to tumorigenicity, immunogenicity, and aberrant cellular proliferation should be monitored further. Pre-clinical research should be used to anticipate such adverse events, and the clinical protocol should include information on toxicity grading and management techniques.

Before enrollment, subjects should be asked to consent voluntarily and knowingly to ongoing monitoring (LTFU). The long-term effects of deliberate and unintentional editing at on- and off-target loci may not have been known when GE products were administered, as was previously mentioned. Developers are advised to perform LTFU at least 15 years after the administration of the product (long-term follow-up).

The 15-year follow-up of patients may not be feasible; the filing should state thus.

## 8. Fast-to-Market Strategies

Gene editing technology development to test its applications is now one of the most straightforward processes, with all components required to be off-the-shelf available. The supplies are available with cGMP certification, qualifying their use to test in humans. Most companies develop their technology or outsource; today, the demand for CDMOs is high, but many new entries are anticipated to overcome this constriction soon. Several companies have built manufacturing facilities to support clinical and commercial manufacturing. Still, even modest facilities require significant capital investment in the tens of millions of dollars and with significant burn rates.

The process of designing and testing gene editing products is well-defined, and technology is widely available [31]. Notably, several educational institutions also offer these services [32], in addition to commercial suppliers.

The FDA and EMA provide details about the regulatory filings and details of biological products, including gene therapy products; these reports should be the starting step to understanding the scope of studies expected by the agencies to reduce the time to approval. For example, a recently approved gene therapy product, ZYNTEGLO, took ten years from filing to approval [33]. A word of caution for developers is to know that the regulatory guidelines provided by the FDA or EMA are neither binding on them nor on the developer. Following the path of similar approved products can often lead to a longer path; the developers must question every requirement before and during the development process.

The fast development of CRISPR tools involves a systematic approach:

■ Tools to generate a complete knock-in design, edit up to 30 bases in any human gene using CRISPR-Cas9 or TALEN technology, design all necessary oligos for precise SNP or amino acid changes, and design the required reagents to add a GFP or RFP tag to a target gene without the need for cloning are now available as off-the-shelf items.

■ The design of the gRNA is essential to the CRISPR-Cas9 system's editing effectiveness, and several proprietary designs are offered for maximum editing effectiveness without sacrificing specificity. For instance, there is transfection of the Cas9 protein and guide-RNA (gRNA) bypass transcription and translation to increase editing efficiency, allowing CRISPR plasmids to remain in the cell for longer than 72 h and potentially contributing to off-target events. In addition, within 24 h, the Cas9 Protein v2 is removed from the cell, reducing the possibility of off-target cleavage events.

■ Quality control is required to improve delivery circumstances, increase editing effectiveness, and establish hit selection criteria. Additionally, these controls aid in developing assays with improved signal-to-noise ratios, resulting in greater assurance in the hits identified by the screens.

■ Nontargeting gRNA sequences that do not recognize any sequences in the human genome serve as negative controls. There are various package sizes for the negative controls. When developing assays and running your screens, negative controls are used to check for non-specific cellular effects on the plate.

■ Validated gRNA sequences showing high editing efficiencies in various cell types—up to 90% in some cell types—are considered positive controls. The conditions that provide the greatest editing efficiency in cell models are identified using these controls during the assay development process. Then, when you run your screens, they can act as on-plate positive controls.

Some Key CRISPR-Cas9 system resources include:

● CasOFFinder is used to predict off-target sites and off-target editing [34].
● Ready-made Cas9 vectors, such as:
  ○ Alt-R CRISPR-Cas9 from Integrated DNA Technologies [35].
  ○ GeneArt CRISPR Nuclease Vector Kit [36].
  ○ LentiCRISPR vector V2 52961 [37].
● Resources for obtaining ZFNs and TALENS for DNA targets include commercial vendors [38].

- Zinc Finger Tools [39] is a public website that lets users look for potential nuclease target sites in a DNA sequence of interest. In addition, this website provides researchers with a database of described zinc finger domains and a reverse engineering feature that forecasts the binding locations of recognized zinc finger proteins.
- Zinc Finger Consortium. Context-dependent Assembly (CoDA), one of the publicly accessible techniques for designing zinc finger domains [40], Oligomerized Pool Engineering (OPEN), and Modular Assembly [41].
- TALENs [42].
- Cellectis Bioresearch and Life Technologies; https://cellecta.com/ (accessed on 8 September 2023).
- Modular assembly Voytas Kit [43], the Zhang kit [44]. These kits produce TALE libraries and arrays using a variety of techniques.
- Library of prefabricated TALE arrays; FLASH [45].
- Library of TALEN plasmids for over 18,000 human protein-coding genes [19].
- Mammalian expression vectors to produce TALENS in just two days methodology [46].

## 9. Cost Containment

While the suggestions made above to reduce the regulatory cost seem practical, there is no assurance that the regulatory agencies will soon be willing to switch their thinking from traditional to creative. As a result, the cost of these products shall remain unaffordable unless covered under insurance plans. Even when the insurance plan picks up most of the cost, the out-of-pocket costs remain formidable. Declaring GE tools suitable for inclusion under the Pasteur Act is one solution to this problem [47]. The developers are paid a one-time reimbursement by the government or a joint fund developed for the purpose; this will allow all patients to receive the treatment free of charge. This proposition applies best to gene therapies where the number of patients is much smaller and, in some cases, predictable [48]. These numbers can range from a dozen to a thousand [49]. This will also allow the developer to estimate their lifetime cost burden.

The approach to bringing "biosimilar" gene therapy products when the GE products are approved will not be applicable; even if it did, the 12-year exclusivity delay would be an impediment. But if it is filed as a new biologic, the exclusivity period will not apply. GE products can capitalize on the concessions given to similar new biologics (approved under 351a).

The FDA's instructions for COVID-19 products are essential: "To the extent that it is both legally and scientifically possible, the development of the COVID-19 vaccine may be expedited using the knowledge obtained from related products made with the same well-characterized platform technology. Similar to how some manufacturing and control processes may draw on the vaccine platform with the right justification, reducing the need for data specific to a given product in some cases." Developers can build a case of safety and efficacy using an approved product or a product for which public domain data are available to secure testing concessions.

Only 27% of gene therapy patents remain valid as of 2022, another crucial aspect [50].

Material costs are often the major hurdle in developing new products, but a revolution regarding GE products has already come to fruition. Off-the-shelf GE kits perform perfectly well and provide a widely available starting point.

Generally, a kit costing less than USD 200 can perform the complete gene editing of a bacteria, including all chemicals, bacteria, media, and glassware. Comparing these costs to create a GE product ready for non-clinical testing is inappropriate, but these costs should be at least a few thousand dollars. The safety and efficacy testing costs would be higher, but a total of USD 100 K to reach nonclinical testing is possible. The cGMP-compliant chemicals and supplies will add a few thousand more, allowing the developers to develop many test products to select the most suitable human use.

## 10. Conclusions

If billions who need it can afford GE, it will revolutionize human history. There is no pharmacology or contact toxicity pattern, unlike biological drugs. RNA-based CRISPR is well-characterized and target-specific. Unlike biological medications, batch variation is low. It resembles chemical medications because its components are well-defined. Off-target genome editing is a concern, yet techniques to quantify it are unreliable. After GMP compliance, regulatory bodies should allow faster testing of these tools, which can only be tested in patients. The off-the-shelf GMP-grade ingredients should allow for the developing of various goods without investing billions. GE tools will need more regulatory reasons and developmental ingenuity to circumvent the price structure that is holding back gene therapies.

**Funding:** This research received no external funding.

**Conflicts of Interest:** The authors declare no conflict of interest.

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
