# Peer review of "Gene Editing: The Regulatory Perspective"

_encyclopedia, doi:10.3390/encyclopedia3040096_

Round 1
Reviewer 1 Report
This entry systemically reviewed the considerations of genome editing technology in drug substance, drug production, preclinical studies, safety and toxicity, and so on, which provides a solid reference for further elucidation and necessary revisions to improve the guidelines about the application of gene editing. It also provides helpful and interesting information for the researchers and the public to learn about the genome editing technology. I think it’s an excellent entry. However, I hope the author could supply some discussion on the ethical issues related to gene editing technology. For example, what does the author think of He Zhengkui's research on using gene editing technology to help AIDS patients rear healthy babies?
Author Response
REVIEWER 1
This entry systemically reviewed the considerations of genome editing technology in drug substance, drug production, preclinical studies, safety and toxicity, and so on, which provides a solid reference for further elucidation and necessary revisions to improve the guidelines about the application of gene editing. It also provides helpful and interesting information for the researchers and the public to learn about the genome editing technology. I think it’s an excellent entry. However, I hope the author could supply some discussion on the ethical issues related to gene editing technology. For example, what does the author think of He Zhengkui's research on using gene editing technology to help AIDS patients rear healthy babies?
RESPONSE:
Thank you very much. Since this topic is big, I had to split it into two papers; the first part has been published :
Niazi, S.K. The Dawn of In Vivo Gene Editing Era: A Revolution in the Making. Biologics 2023, 3, 253-295. https://doi.org/10.3390/biologics3040014
https://www.mdpi.com/2673-8449/3/4/14
In this paper I have discussed in detail the ethical issues including the subject you had mentioned above. In this paper, I focused only on the regulatory aspect for securing approval. I hope you find it acceptable.
Reviewer 2 Report
The paper is devoted to the discussion of regulation of the creation and application of genome editing and gene therapy methods. These methods are currently undergoing extremely rapid development. Undoubtedly, the use of such a new approach should be adequately controlled, and the legislation should correspond to the level of development of these technologies. The author points out the specificity of this therapy, which requires special approaches to regulate its use. The author points out a number of shortcomings in the FDA guidance on the use of these methods.
The publication discusses various requirements for gene-therapeutic drugs, the necessary stages of their testing, issues of toxicity and possible side effects, issues of preclinical and clinical studies, peculiarities of market launch and pricing. In general, the publication provides a comprehensive view on various practical aspects of the development, testing and promotion of genotherapeutic medicines and provides a good basis for the further development of regulatory documents in this area. The publication will be useful for experts in this field.
There are a few minor typos in the text that require further careful checking.
Author Response
REVIEWER 2
The paper is devoted to the discussion of regulation of the creation and application of genome editing and gene therapy methods. These methods are currently undergoing extremely rapid development. Undoubtedly, the use of such a new approach should be adequately controlled, and the legislation should correspond to the level of development of these technologies. The author points out the specificity of this therapy, which requires special approaches to regulate its use. The author points out a number of shortcomings in the FDA guidance on the use of these methods.
The publication discusses various requirements for gene-therapeutic drugs, the necessary stages of their testing, issues of toxicity and possible side effects, issues of preclinical and clinical studies, peculiarities of market launch and pricing. In general, the publication provides a comprehensive view on various practical aspects of the development, testing and promotion of genotherapeutic medicines and provides a good basis for the further development of regulatory documents in this area. The publication will be useful for experts in this field.
There are a few minor typos in the text that require further careful checking.
RESPONSE: I have gone through complete Grammarly check to fix these errors; if accepted, I will secure professional editing to ensure full compliance with your comments.
Reviewer 3 Report
In the manuscript entitled “Gene Editing: The Regulatory Perspective”, the author aimed to analyzie gene editing (GE) in its future regulatory perspectives.
He made a deep dissection about gene or genome editing (GE) and gene therapy (GT) and reported the state-of-approval regarding GE, as well as, GT products.
The manuscript is clear and well written, but there are two main issues that remains to be covered. The manuscript lacks of a great number of references for all the comments and examples the author reported. Please add them within the manuscript since in this form it is difficult to check for all the informations the author reported in.
It would be interesting and very informative to add some graphical images regarding the experimental protocol used in most of the treatments and workflows commented.
The author should also add an ABBREVIATIONS Section to the manuscript in order to clarify the meaning of such acronyms reported within the text.
It is reported within the text “Figure 2”, but I hadn’t received any figure. At the same time it is not reported “Figure 1” before the “Figure 2”. Please check fot them within the text and sent files.
With regards to Table 1, I think it is good to have an example of suppliers that are involved in GE activities but it has to be taken as a representative list of references for this kind of therapies.
In conclusion, I think that in this paper, the author commented in an interesting way the state of the art for gene or genome editing (GE) and gene therapy (GT) but the manuscript needs to be implemented based on comments I reported above. This manuscript can’t be taken in consideration for publication in its present form but may be considered after the author responded to my comments.
No comments
Author Response
REVIEWER 3
In the manuscript entitled “Gene Editing: The Regulatory Perspective”, the author aimed to analyzie gene editing (GE) in its future regulatory perspectives.
He made a deep dissection about gene or genome editing (GE) and gene therapy (GT) and reported the state-of-approval regarding GE, as well as, GT products.
The manuscript is clear and well written, but there are two main issues that remains to be covered. The manuscript lacks of a great number of references for all the comments and examples the author reported. Please add them within the manuscript since in this form it is difficult to check for all the informations the author reported in.
It would be interesting and very informative to add some graphical images regarding the experimental protocol used in most of the treatments and workflows commented.
The author should also add an ABBREVIATIONS Section to the manuscript in order to clarify the meaning of such acronyms reported within the text.
It is reported within the text “Figure 2”, but I hadn’t received any figure. At the same time it is not reported “Figure 1” before the “Figure 2”. Please check fot them within the text and sent files.
With regards to Table 1, I think it is good to have an example of suppliers that are involved in GE activities but it has to be taken as a representative list of references for this kind of therapies.
In conclusion, I think that in this paper, the author commented in an interesting way the state of the art for gene or genome editing (GE) and gene therapy (GT) but the manuscript needs to be implemented based on comments I reported above. This manuscript can’t be taken in consideration for publication in its present form but may be considered after the author responded to my comments.
RESPONSE: Thank you very much. I have fixed the mistakes in the listing of figures as you have pointed out; also in compliance with the journal format, I have made sure that an abbreviation is identified when it is listed for the first time. Regarding the GE suppliers, I have worked on that in my first paper on the topic; this division was necessary because of the length of the paper. Here is the one published earlier that includes all details you have suggested.
Niazi, S.K. The Dawn of In Vivo Gene Editing Era: A Revolution in the Making. Biologics 2023, 3, 253-295. https://doi.org/10.3390/biologics3040014
https://www.mdpi.com/2673-8449/3/4/14
Round 2
Reviewer 1 Report
I am satisfied with the revised paper.
Reviewer 2 Report
The author has replied to the issue raised.
Reviewer 3 Report
No comment